# Phenotypic and Genetic Evidence for a More Prominent Role of Blood Glucose than Cholesterol in Atherosclerosis of Hyperlipidemic Mice

**DOI:** 10.3390/cells11172669

**Published:** 2022-08-28

**Authors:** Ashley M. Abramson, Lisa J. Shi, Rebecca N. Lee, Mei-Hua Chen, Weibin Shi

**Affiliations:** Department of Radiology and Medical Imaging, University of Virginia, Charlottesville, VA 22908, USA

**Keywords:** atherosclerosis, dyslipidemia, hyperglycemia, quantitative trait locus, type 2 diabetes

## Abstract

Hyperlipidemia and type 2 diabetes (T2D) are major risk factors for atherosclerosis. *Apoe*-deficient (*Apoe*^−/−^) mice on certain genetic backgrounds develop hyperlipidemia, atherosclerosis, and T2D when fed a Western diet. Here, we sought to dissect phenotypic and genetic relationships of blood lipids and glucose with atherosclerotic plaque formation when the vasculature is exposed to high levels of cholesterol and glucose. Male F2 mice were generated from LP/J and BALB/cJ *Apoe*^−/−^ mice and fed a Western diet for 12 weeks. Three significant QTL *Ath51*, *Ath52* and *Ath53* on chromosomes (Chr) 3 and 15 were mapped for atherosclerotic lesions. *Ath52* on proximal Chr15 overlapped with QTL for plasma glucose, non-HDL cholesterol, and triglyceride. Atherosclerotic lesion sizes showed significant correlations with fasting, non-fasting glucose, non-fasting triglyceride, and body weight but no correlation with HDL, non-HDL cholesterol, and fasting triglyceride levels. *Ath52* for atherosclerosis was down-graded from significant to suggestive level after adjustment for fasting, non-fasting glucose, and non-fasting triglyceride but minimally affected by HDL, non-HDL cholesterol, and fasting triglyceride. Adjustment for body weight suppressed *Ath52* but elevated *Ath53* on distal Chr15. These results demonstrate phenotypic and genetic connections of blood glucose and triglyceride with atherosclerosis, and suggest a more prominent role for blood glucose than cholesterol in atherosclerotic plaque formation of hyperlipidemic mice.

## 1. Introduction

Atherosclerosis and type 2 diabetes (T2D) are two major health problems worldwide. The former is the pathological cause of coronary heart disease, ischemic stroke, and peripheral arterial disease. The latter is a chronic metabolic disorder featured by chronic hyperglycemia that results from concomitant insulin sensitivity and beta-cell impairment [1]. The two disorders share many risk factors, including high-fat and carbohydrate diet, sedentary lifestyle, obesity or overweight, older age, smoking, air pollution, hypertension, and dyslipidemia [2,3], and tend to co-occur [4]. Their comorbidity has led to speculation that the two disorders share genetic factors and pathways [4]. A few rare gene mutations result in both early coronary heart disease and T2D that are observable as Mendelian traits segregating in families, and these genes include *LRP6* [5], *ABCA1* [6,7], *LIPE* [8], *LPL* [9], and *APOB* [10]. The common forms of atherosclerosis and T2D involve many genes and exhibit significant gene–environment interactions. Human genome-wide association studies (GWAS) have shown positive genetic correlations between coronary heart disease and T2D, but few shared loci have been found [11]. The major hurdles to such studies in humans include small effect sizes of the loci, genetic heterogeneity, and environment variation. Mouse models provide a unique complimentary tool as genetic heterogeneity can be minimized by using inbred strains and environmental factors be strictly controlled. Moreover, when a locus for a complex trait or disease is found, genetic and genomic tools are available for confirming its existence, narrowing down its confidence interval, and validating underlying causal genes.

*Apoe* null (*Apoe*^−/−^) mice develop spontaneous hyperlipidemia and atherosclerosis on a chow diet, which are accelerated on a high fat diet [12]. We have found that *Apoe*^−/−^ mice on certain genetic backgrounds, such as C57BL/6, C3H/HeJ, and SWR/J, develop T2D (fasting plasma glucose >250 mg/dL) after a prolonged exposure to a Western diet [13,14]. In multiple intercrosses derived from *Apoe*^−/−^ mouse strains, we have observed colocalization of QTL for atherosclerotic lesions with QTL for blood glucose [15,16,17]. The colocalization of QTL for two different traits in the confidence interval provides a genetic means for inferring causal connections between them. This analysis is based on the speculation that if two traits are causally linked, phenotypical variation in one trait will affect the other; otherwise unlikely [18]. BALB/cJ (BALB) and LP/J (LP) are among the 16 mouse strains whose genomes have been sequenced and sequence variant data are publicly accessible [19]. The two strains display variations in atherosclerosis susceptibility and metabolic traits [20,21]. In this study, we constructed a segregating F2 population from BALB and LP *Apoe*^−/−^ mice to partition phenotypic and genetic connections between atherosclerosis and T2D.

## 2. Materials and Methods

### 2.1. Mice

LP and BALB *Apoe*^−/−^ mouse strains were created in our laboratory as reported [14]. F2 mice were generated from an intercross between male LP-*Apoe*^−/−^ mice and female BALB-*Apoe*^−/−^ mice. Male mice were fed 12 weeks of Western diet containing 21% fat, 34.1% sucrose, 0.15% cholesterol, and 19.5% casein (TD 88137, Envigo, Dublin, VA, USA), starting at 6 weeks of age. Both male and female *Apoe*^−/−^ mice develop atherosclerosis and T2D on the Western diet [22,23,24]. Our current focus on male mice reduced workload by half when compared to the design involving both sexes of mice. Mice were bled three times: once before the start of the Western diet, once without fasting at the 11th week of Western diet, and once after overnight fast at the end of the Western diet. Retro-orbital venous blood was drawn when mice were anesthetized through isoflurane inhalation and prepared as described [16]. All procedures were performed according to an animal protocol approved by the Institutional Animal Care and Use Committee (protocol #: 3109).

### 2.2. Measurement of Atherosclerotic Lesions

Atherosclerotic lesions in the aortic root were quantified as reported with minor modifications [15,25]. In brief, the vasculature of mice was first flushed with saline and then perfused with 10% formalin through the left ventricle of the heart. The aortic root and adjacent heart were dissected, embedded in a cutting medium (Tissue-Tek, Torrance, CA, USA), and cut in 10-μm thickness. Frozen sections were stained with oil red O and hematoxylin, and atherosclerotic lesions were measured using Zeiss AxioVision 4.8 software. Lesion areas on five sections with the largest readings were averaged for each mouse, and this value was used for QTL mapping.

### 2.3. Glucose Assay

Plasma concentrations of glucose were measured using a Sigma kit (Cat. # GAHK20) as reported [26]. Briefly, 6 µL diluted plasma samples (3× in H_2_O), standards, and controls were added in a 96-well plate, incubated with assay reagent for 30 min at 30 °C, and absorbance at 340 nm measured using a Molecular Devices plate reader.

### 2.4. Lipid Assays

Plasma levels of total, HDL, non-HDL cholesterol, and triglyceride were measured by enzymatic assays as reported [26,27].

### 2.5. Genotyping

DNA was extracted from tails of mice using a QIAGEN DNeasy kit. Genotyping was done at Neogen (Lansing, MI, USA) with miniMUGA SNP arrays harboring 11,000 probes. Controls (parental and F1 DNA) were included for each array. Uninformative SNP markers as well as informative ones that did not show the expected genotypes for control samples were excluded. We checked genotyping errors using the “calc errorlod” function of R/qtl. 2611 SNP markers passed the test and were used for QTL analysis.

### 2.6. Statistical Analysis

QTL analysis was carried out using R/qtl and Map Manager QTX as reported [17,28,29]. 1000 permutation tests were performed to define genome-wide LOD (logarithm of odds) score thresholds for significant and suggestive QTL with each phenotype. QTL with a genome-wide *p* value < 0.05 were significant, and those with a genome-wide *p* value < 0.63 were suggestive [30]. For interval mapping using QTX, redundant markers were excluded so that every marker had a unique genotyping result for the entire F2 cross.

### 2.7. Causal Inference from Deep Analysis of Overlap QTL

When coincident QTL for two complex traits were found, further analysis was performed to infer likely causal relationships between them as described [17,31]. Residuals resulting from linear regression analysis of two related traits were subject to genome-wide QTL scan as a new phenotype. The QTL detected from residual variation in one trait would be independent of variation in the other.

### 2.8. Prioritization of Candidate Genes

We used bioinformatics resources to prioritize candidate genes for significant atherosclerosis QTL. Genetic variants among mouse strains were searched using the Sanger Mouse Genomes Project (https://www.sanger.ac.uk/sanger/Mouse_SnpViewer/rel-1505, accessed on 4 May 2022). Those genes containing one or more missense SNPs or SNP(s) in upstream regulatory regions within the confidence interval of a QTL were considered probable candidates as described [32,33,34]. SIFT (Sorting Intolerant From Tolerant) score was queried through the Ensembl Genome Browser (https://useast.ensembl.org/index.html, accessed on 4 May 2022) and used to estimate the impact of a nonsynonymous variant on protein function [35].

## 3. Results

### 3.1. Atherosclerotic Lesion Analysis

Atherosclerotic lesion areas in the aortic root were measured after male F2 mice were fed the Western diet for 12 weeks. Cryosections were stained with oil red O and hematoxylin (Figure 1). The F2 mice exhibited a wide range lesion distribution with a 26-fold difference between the mouse having the largest lesion (361,448 µm^2^/section) and the mouse having the smallest lesion (14,140 μm^2^/section). The lesion area values of F2 mice approached a normal distribution.

### 3.2. QTL Analysis

A genome-wide scan revealed three significant QTL on Chr3 and Chr15 and one suggestive QTL on Chr13 for atherosclerotic lesion sizes (Figure 2A). Details of these QTL, such as locus name, LOD score, peak marker, confidence interval, high allele, mode of inheritance, and allelic effect are showed in Table 1. The Chr3 QTL, named *Ath51*, had a significant LOD score of 4.35 and peaked at 94.5 Mb. The BALB allele increased lesion size and the LP allele decreased lesion size at the locus (Table 1). This QTL is partially overlapping with *Ascla4* (55 cM or 110 Mb), previously mapped in B6 × FVB *Ldlr*^−/−^ F2 mice [36].

LOD score plot for Chr15 revealed two distinct peaks, indicating the existence of two QTL for atherosclerosis (Figure 2A,B): The proximal QTL, named *Ath52*, had a significant LOD score of 5.42 and peaked at 43.1 Mb, and the distal QTL, named *Ath53*, had a significant LOD score of 4.17 and peaked at 99.0 Mb. For both QTL, the LP allele was associated with larger lesion size and the BALB allele associated with smaller lesion size (Table 1). A significant (*Ath33*) and a suggestive atherosclerosis QTL (*Ath23*) were mapped to Chr15: 79 cM (~158 Mb) and Chr15: 68 Mb from B × H and AKR × DBA/2 *Apoe*^−/−^ F2 intercrosses [37,38], but neither is coincident with the QTL we mapped in this study.

A suggestive QTL for atherosclerosis was mapped to Chr13: 103.6 Mb and had a LOD score of 2.74 (Figure 2A,B). The LP allele was associated with larger lesion size. This QTL replicates *Ath32* mapped in a B × H *Apoe*^−/−^ F2 intercross [38].

### 3.3. Coincident QTL for Atherosclerotic Lesions, Plasma Glucose and Lipids

Interval mapping plots for Chr15 show the coincidence of the proximal atherosclerosis QTL (*Ath52*) with QTL for fasting, non-fasting plasma glucose (*Bglu20*), LDL cholesterol and triglyceride levels (*Nhdlq18*) (Figure 3A–G). Details on *Bglu20* and *Nhdlq18* and the characterization of plasma lipids and glucose in the F2 mice have been reported [16,39]. The LP allele was associated with larger atherosclerotic lesion size and higher plasma levels of glucose, non-HDL cholesterol and triglyceride, while the BALB allele had opposite effects on these traits (Table 1).

### 3.4. Associations of Atherosclerotic Lesion Sizes with Plasma Glucose, Lipid Levels and Body Weight

Atherosclerotic lesion sizes showed significant correlations with plasma levels of fasting (r = 0.203; *p* = 0.015), non-fasting glucose (r = 0.165; *p* = 0.049), non-fasting triglyceride (r = 0.20; *p* = 0.016), and body weight (r = −0.236; *p* = 0.004) (Figure 4A–I). A trend toward correlation was observed with fasting triglyceride (r = 0.115; *p* = 0.173) and non-fasting non-HDL cholesterol levels (r = 0.116; *p* = 0.168). No correlation was found with fasting non-HDL (r = 0.116; *p* = 0.168), fasting (r = 0.080; *p* = 0.34) and non-fasting HDL cholesterol levels (r = 0.080; *p* = 0.34).

### 3.5. Causal Associations of Atherosclerotic Lesion Sizes with Plasma Glucose, Triglyceride and Body Weight

As the QTL for atherosclerotic lesions (*Ath52*) was coincident with QTL for plasma glucose (*Bglu20*) and lipid levels (*Nhdlq18*) on Chr15, we examined their potential causal relationships. Residuals from regression analysis of atherosclerotic lesion sizes with each of the fasting, non-fasting glucose and lipid traits were analyzed to find QTL contributing to residual variation in atherosclerotic lesion sizes. When residuals from analysis with fasting or non-fasting glucose were analyzed, both Chr15 QTL for atherosclerosis showed reductions in LOD score (proximal QTL: 4.0, distal QTL: 3.5 for fasting glucose; proximal QTL: 3.75, distal QTL: 3.72 for non-fasting glucose) (Figure 5B,C), implying a causal role for blood glucose in atherosclerotic lesion formation.

When residuals from regression analysis with fasting non-HDL cholesterol level were analyzed, both Chr15 QTL for atherosclerosis showed little change in their LOD scores (proximal QTL: 5.28, distal QTL: 4.04) (Figure 5D). After adjustment for non-fasting LDL, both Chr15 QTL showed a slight reduction in LOD score (proximal QTL: 4.31, distal QTL: 3.8) (Figure 5E).

The proximal Chr15 QTL for atherosclerosis showed a slight reduction in LOD score (4.79) after adjusting fasting triglyceride level but a noticeable reduction (4.03) after adjusting non-fasting triglyceride level (Figure 5F,G). Distal Chr15 QTL showed little change in LOD score after adjustment of fasting (3.9) or non-fasting triglyceride levels (4.0).

Both atherosclerosis QTL on Chr15 showed little change in LOD score after adjustment of fasting or non-fasting HDL cholesterol levels (Figure 5H,I).

After adjustment of body weight, the proximal Chr15 QTL for atherosclerosis showed a reduced LOD score (4.3) but the distal QTL had an increased LOD score (4.82) (Figure 5J).

To determine the influence of atherosclerotic lesions on plasma glucose levels, we performed genome-wide scans on residuals from regression analysis of glucose versus atherosclerotic lesions. After adjustment for atherosclerotic lesions, the peak LOD score of the Chr15 QTL for fasting glucose dropped from a significant (5.36) to suggestive level (3.72). For nonfasting glucose, the peak LOD value of the Chr15 QTL dropped from 7.03 to 5.21 (Figure 6), suggesting the existence of genetic connection between atherosclerosis and T2D on the chromosome.

### 3.6. Prioritization of Candidate Genes for Significant QTL

A significant QTL for atherosclerosis on Chr3 near *Ath51* has been mapped in B6 × FVB *Ldlr*^−/−^ F2 mice [36]. At the QTL, the BALB and B6 alleles increased atherosclerosis whereas the LP and FVB alleles reduced it. Over 30 genes within the confidence interval of *Ath51* (89.8–121.8 Mb) contain one or more missense SNPs or SNP(s) in upstream regulatory regions that are shared by high allele strains but different from SNPs shared by low allele strains (Table 2). Of them, *Hrnr*, *Tchhl1*, *Fcgr1*, *Hsd3b2*, *Wars2*, *Mab21l3*, and *Bcas2* contain one or more missense variants with a low SIFT score predicted to affect protein function.

For proximal Chr15 QTL (*Ath52*), probable candidate genes include *Erich5*, *Vps13b*, *Rgs22*, *Fbxo43*, *Snx31*, *Baalc*, *Oxr1*, *Tmem74*, *Pkhd1l1*, *Sybu*, and *Trmt12* (Appendix A). *Endou*, *Hdac7*, *Senp1*, *Olfr286*, *Olfr285*, *Nckap5l*, *Smarcd1*, *Lima1*, *Fam186a*, and *Espl1* are probable candidate genes for distal Chr15 atherosclerosis QTL (*Ath53*). These positional candidate genes contain one or more missense SNPs that have low SIFT scores predicted to affect protein function in either of the two parental strains.

## 4. Discussion

In the present study, we have identified three significant QTL on chromosomes 3 and 15 for atherosclerosis and observed overlapping of the QTL for aortic atherosclerosis with those for non-HDL cholesterol, triglyceride, and glucose on Chr15 in male F2 mice derived from two *Apoe*^−/−^ mouse strains. The proximal QTL for atherosclerosis on Chr15 was down-graded after eliminating variation in fasting, non-fasting glucose, non-fasting triglyceride levels, or body weight but was minimally affected by eliminating variation in fasting, non-fasting HDL and non-HDL cholesterol levels. Moreover, atherosclerotic lesion sizes showed significant correlations with plasma levels of fasting, non-fasting glucose, non-fasting triglyceride, and body weight but not with fasting, non-fasting HDL, and non-HDL cholesterol levels among the F2 population.

We identified a significant QTL *Ath51* for atherosclerosis on chromosome 3 at 94.5 Mb that partially overlaps in the confidence interval with *Ascla4* (55 cM or 110 Mb), a significant atherosclerosis QTL previously mapped in a B6 × FVB *Ldlr*^−/−^ intercross [36]. In that intercross, a significant atherosclerosis QTL, *Ascla3*, was also mapped to distal chromosome 3 region at 79 cM or 160 Mb. A suggestive atherosclerosis QTL, *Ath23*, was mapped to a more proximal region (68 Mb) than *Ath51* in an AKR × DBA/2 *Apoe*^−/−^ F2 cross [37]. As *Ath51* and *Ascla4* have been mapped in two crosses derived from different parental strains, we were able to use DNA sequence variant data to prioritize positional candidate genes containing one or more missense SNPs or SNP(s) in upstream regions that segregate between the high allele strains (BALB, B6) and low allele strains (LP, FVB). Since 97% of the genetic variants between common mouse strains are ancestral [40], it is almost certain that QTL genes are those that contain polymorphisms shared among mouse strains. *Hrnr*, *Tchhl1*, *Fcgr1*, *Hsd3b2*, *Wars2*, *Mab21l3*, and *Bcas2* were identified as top candidate genes, with each containing one or more missense SNPs of low SIFT scores that are predicted to impact protein function. *Fcgr1* deficiency has shown protection against atherosclerosis in *Apoe*^−/−^ mice [41].

We identified two significant QTL for atherosclerosis on chromosome 15 at 43.1 and 99 Mb. Previously, two significant atherosclerosis QTL, *Ath22* and *Ath33*, have been mapped to chromosome 15 at 20 and 71 Mb using an AKR × DBA/2 and two B × H *Apoe*^−/−^ F2 intercrosses [37,38,42]. However, neither was coincident with the atherosclerosis QTL, *Ath52* and *Ath53* that we had identified in this study.

An intriguing finding of this study was the coincidence of atherosclerosis QTL (*Ath52*) with QTL for plasma glucose (*Bglu20*), non-HDL cholesterol (*Nhdlq18*), and triglyceride on chromosome 15 in the current cross. This enables for elucidating the likely causal relationship between the traits. Using a causal inference test through excluding the influence of variation in plasma glucose, non-HDL cholesterol, or triglyceride levels, we were able to demonstrate the dependence of the atherosclerosis QTL on plasma glucose and triglyceride but not on non-HDL cholesterol levels. Indeed, after adjustment for fasting or non-fasting glucose, both Chr15 atherosclerosis QTL were downgraded from significant to suggestive ones. Thus, the current findings offer genetic evidence that hyperglycemia plays a causal role in the development of atherosclerosis.

We found that the peak LOD score of the Chr15 QTL for either fasting or non-fasting glucose levels (*Bglu20*) dropped after adjustment for atherosclerotic lesion areas. The bi-direction of genetic effects from the Chr15 QTL further supports the existence of genetic connection between atherosclerosis and glucose metabolism. Atherosclerotic plaques may have little influence on glucose hemostasis, but concomitant inflammation developed on a high fat diet increases insulin resistance and impairs beta-cell function [14,43]. In addition, mice fed a high-fat diet display marked increases in tissue resident macrophages, which uptake and consume more glucose than many normal tissues and organs [44].

Plasma insulin was not measured for this cross; thus HOMA-IR (Homeostatic Model Assessment for Insulin Resistance) and the role of insulin resistance in the etiology of type 2 diabetes and atherosclerosis were unable to be determined. However, we previously demonstrated that impaired insulin secretion rater than insulin resistance underlies the development of Western diet-induced type 2 diabetes in *Apoe*^−/−^ mouse strains [14,43]. Diabetes is defined by fasting hyperglycemia while plasma insulin levels were found poorly correlated with plasma glucose levels in a segregating F2 population derived from *Apoe*^−/−^ mouse strains that developed type 2 diabetes on the Western diet [13].

The proximal but not distal Chr15 atherosclerosis QTL was downgraded from significant to suggestive after adjustment for non-fasting triglyceride levels. In contrast, the influence of fasting triglyceride on the Chr15 atherosclerosis QTL was much smaller. This finding lends support to the notion that non-fasting lipid profile is superior to fasting in predicting cardiovascular risk [45]. The proximal Chr15 atherosclerosis QTL was only slightly down graded after adjustment for non-fasting but not fasting non-HDL cholesterol levels. This finding is unexpected with regard to the crucial role of LDL in foam cell formation but is consistent with the notion that lipid levels may not accurately reflect the numbers of LDL particles, a measure that provides a better estimate of atherogenicity and cardiovascular events [46,47].

The causal connection of QTL effects within the Chr15 region is parallel to the correlations that we found between the affected traits. Indeed, we had observed causal connections of Chr15 atherosclerosis QTL with QTL for fasting, non-fasting glucose and non-fasting triglyceride, so observed were the significant correlations of atherosclerotic lesion sizes with the traits. On the other hand, the Chr15 atherosclerosis QTL had shown no causal connection with QTL for fasting, non-fasting non-HDL and fasting triglyceride, so no correlations of atherosclerotic lesion sizes were observed with these traits.

In this study, we found that atherosclerotic lesion sizes were poorly correlated with plasma levels of HDL, non-HDL cholesterol, and triglyceride among the F2 mice with hyperlipidemia. Similar findings have also been reported in other F2 crosses with hyperlipidemia [28,34,48]. As above discussed, lipid levels are unable to accurately reflect the numbers of LDL particles, which are a better measure of atherogenicity. Another possible explanation for our findings is that all F2 mice have severe hyperlipidemia that exceeds the levels necessary for atherogenesis and thus, any further increase may not contribute to plaque growth.

A significant inverse correlation between atherosclerotic lesion sizes and body weight was observed in F2 mice from the current cross. This inverse correlation is likely due to genetic loci harboring variants that have opposing effects on atherosclerosis and body weight. Indeed, the distal Chr15 atherosclerosis QTL (*Ath53*) was elevated and the proximal QTL (*Ath52*) was suppressed after adjustment for body weight. In female F2 mice from the current cross, we identified a significant QTL on Chr15 for body weight that was coincident to *Ath52* [49]. Although no significant or suggestive QTL for body weight was mapped to the Chr15 region harboring *Ath52* in the male F2 mice, multiple markers with LOD scores above 2.0 were detected for body weight. Body length was not measured for the current cross and thus body mass index (BMI) could not be calculated. Nevertheless, our previous study of a congenic strain and its background strain with magnetic resonance imaging has shown that visceral and subcutaneous fat rather than other tissues underlies the variation in body weight [50]. Visceral and subcutaneous fat is a major contributor to insulin resistance and low-grade systemic inflammation, both of which play a role in type 2 diabetes and atherosclerosis.

In conclusion, the present study is the first to demonstrate the causal relationship between T2D risk and atherosclerosis in a cohort of dyslipidemic F2 mice. Phenotypically, atherosclerotic lesion sizes were significantly correlated with plasma glucose but not HDL and non-HDL cholesterol levels under both fasting and non-fasting conditions. Genetically, Chr15 QTL for atherosclerosis was coincident with QTL for plasma glucose and non-HDL cholesterol levels and suppressed after adjustments for glucose but not HDL and non-HDL cholesterol levels. These findings suggest a more prominent role for glucose than cholesterol in atherosclerotic plaque formation of hyperlipidemic mice. Thus, there is a clear medical need for glucose control besides lipid-lowering therapy for patients with coronary heart disease who have both hyperglycemia and hypercholesterolemia.

## Figures and Tables

**Figure 1 cells-11-02669-f001:**
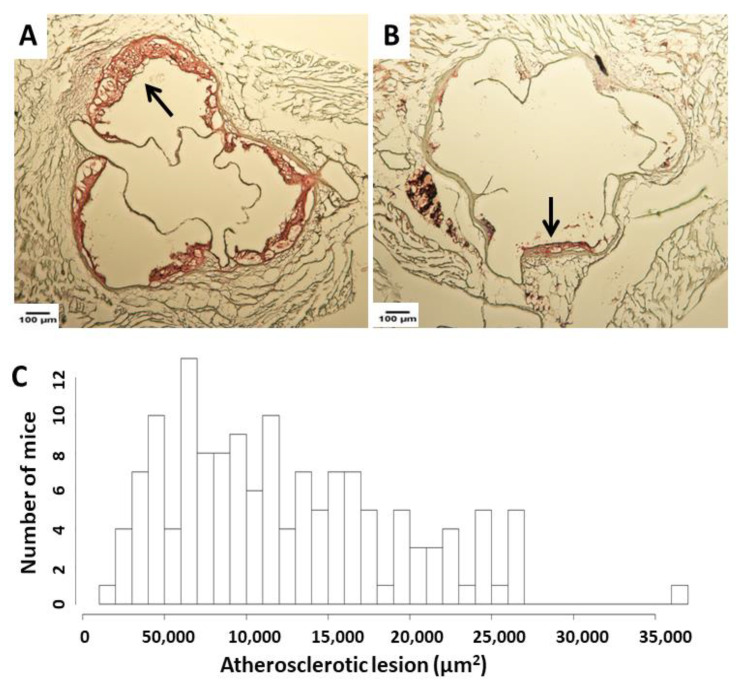
Atherosclerotic lesion analysis in male F2 mice fed 12 weeks of Western diet. (**A**,**B**): Representative images from 2 individual mice. Cross-sections of the aortic root were stained with oil red O and hematoxylin. Arrows point at lesions. (**C**): Atherosclerotic lesion area distribution among F2 mice. The x axis: Lesion areas in µm^2^.

**Figure 2 cells-11-02669-f002:**
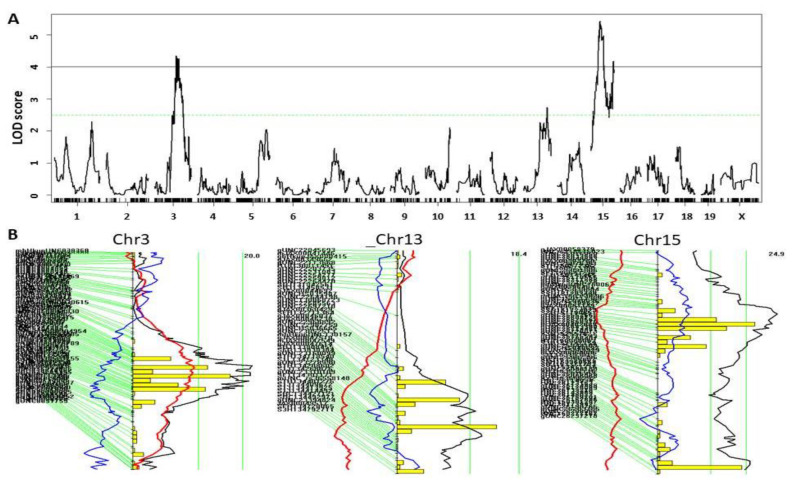
QTL analysis of atherosclerotic lesion sizes in the aortic root of male F2 mice. (**A**) Genome-wide scan to identify loci for atherosclerotic lesions. The x-axis shows the chromosomal position and the y-axis shows the LOD score. Each short vertical bar on the x chromosome represents a marker. Two horizontal lines represent the genome-wide thresholds for *p* = 0.05 (black) and *p* = 0.63 (green). (**B**) Interval mapping plots for chromosomes 3, 13 and 15 harboring atherosclerosis QTL. The black line denotes LOD score, the blue and red lines, respectively, denote dominant and additive regression coefficients. Yellow histograms denote the confidence interval of QTL calculated by the bootstrap test. Two vertical green lines represent genome-wide significance thresholds at *p* = 0.63 and *p* = 0.05 from left to right. Genetic markers are shown on the left of each plot.

**Figure 3 cells-11-02669-f003:**
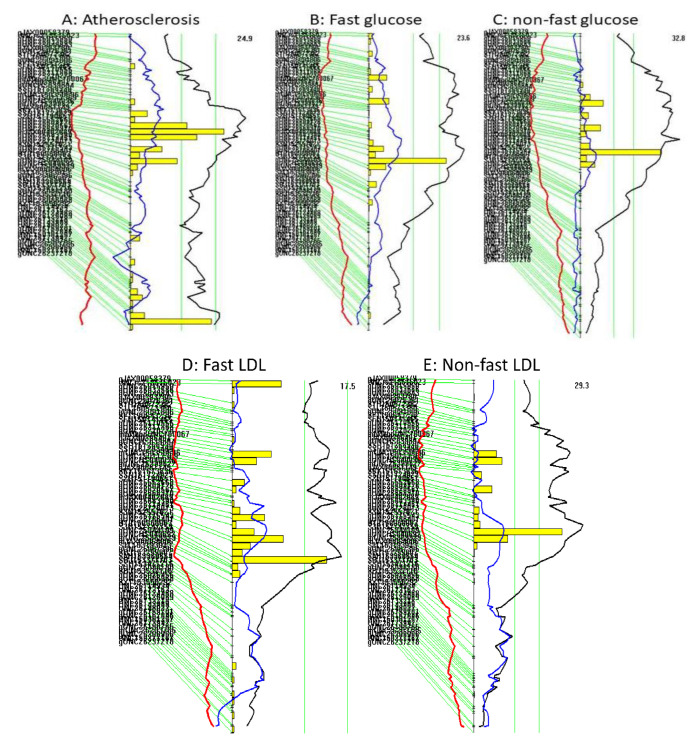
Interval mapping plots show the overlapping of QTL for atherosclerosis (**A**), fasting plasma glucose (**B**), non-fasting glucose (**C**), fasting LDL (**D**), non-fasting LDL cholesterol (**E**), fasting triglyceride (**F**), and non-fasting triglyceride (**G**) on chromosome 15. Plots were created using the interval mapping function of Map Manager QTX. The yellow histograms denote confidence intervals estimated through the bootstrap test. Two vertical green lines denote genome-wide significance thresholds at *p* = 0.63 and *p* = 0.05, respectively.

**Figure 4 cells-11-02669-f004:**
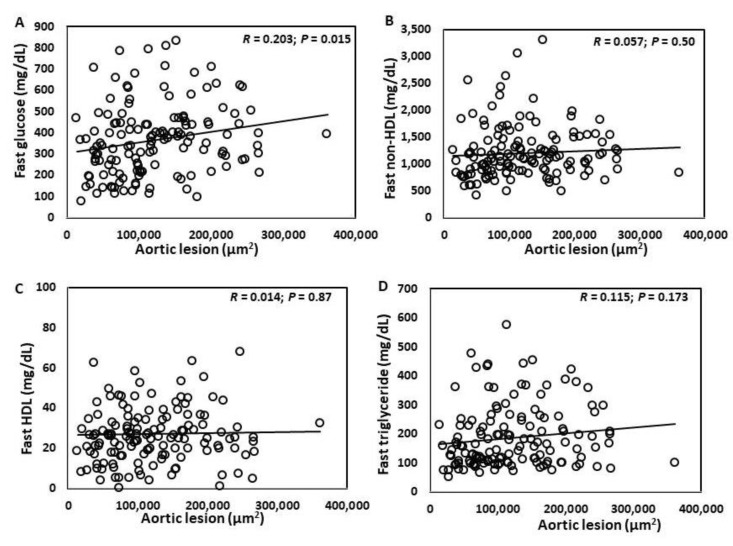
Regression analysis of associations of atherosclerotic lesion sizes with fasting plasma glucose (**A**), non-HDL (**B**), HDL cholesterol (**C**), triglyceride (**D**), non-fasting glucose (**E**), non-HDL (**F**), HDL cholesterol (**G**), triglyceride (**H**), and body weight (**I**) among male F2 mice. Each symbol represents values of a F2 mouse. Correlation coefficient (r) and significance (*p*) values are shown in each figure.

**Figure 5 cells-11-02669-f005:**
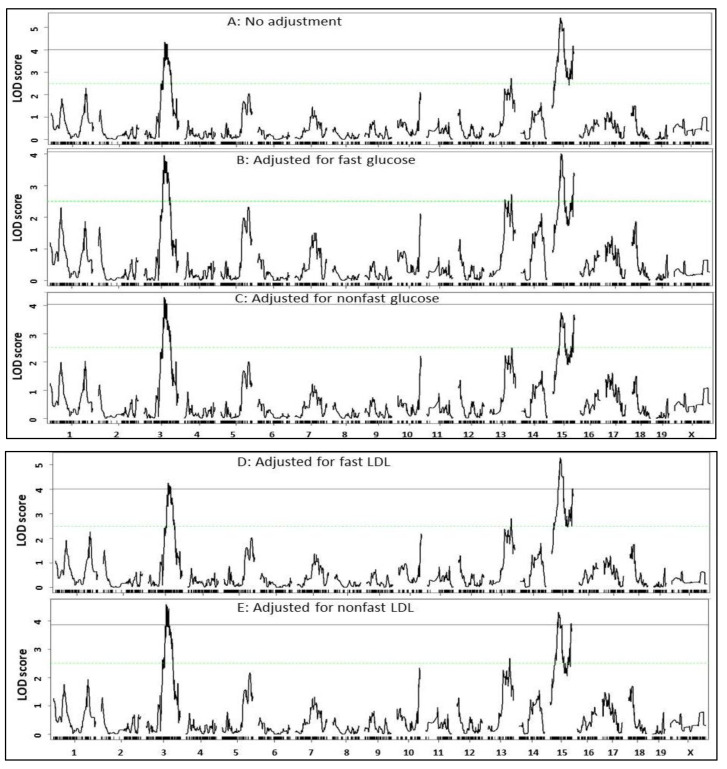
Whole genome scans to evaluate dependence of atherosclerosis QTL (**A**) on fasting glucose (**B**), non-fasting glucose (**C**), fasting LDL (**D**), non-fasting LDL (**E**), fasting triglyceride (**F**), non-fasting triglyceride (**G**), fasting HDL (**H**), non-fasting HDL (**I**), and body weight (**J**). Residuals from regression analysis of atherosclerotic lesion sizes with each measure were analyzed. Note the reduced magnitude of the Chr15 QTL for atherosclerosis after adjustment for fasting, non-fasting glucose, and non-fasting triglyceride, but not fasting, non-fasting LDL, and fasting triglyceride.

**Figure 6 cells-11-02669-f006:**
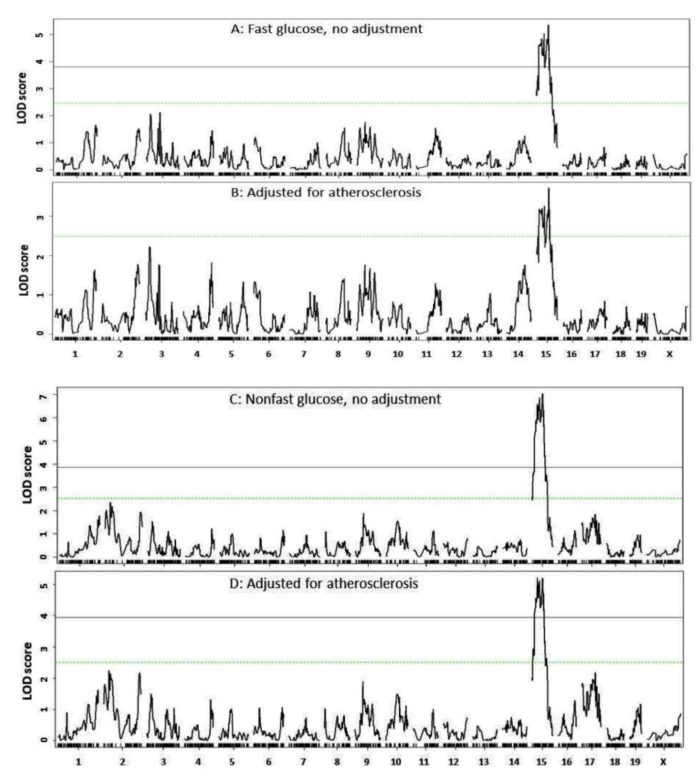
Genome-wide scans to evaluate the influence of atherosclerotic lesions on QTL for fasting glucose (**B**) and non-fasting glucose (**D**). (**A**) Genome-wide scan for fasting glucose without adjustment; (**C**), genome-wide scan for nonfasting glucose without adjustment. Note the peak LOD score drop of Chr15 QTL for fasting and nonfasting glucose levels after adjustment for atherosclerotic lesions.

**Table 1 cells-11-02669-t001:** Significant and suggestive QTL for atherosclerotic lesion sizes mapped with male F2 mice derived from LP and BALB *Apoe*^−/−^ mice.

Locus Name	Chr	LOD ^a^	Peak (Mb)	Closest Marker	95%CI (Mb) ^b^	High Allele	Mode of Inheritance	Allelic Effect ^c^		
								BB	H	LL
** *Ath51* **	3	**4.35**	94.5	gUNC5773934	89.8–121.8	BB	Additive	165964 ± 67352	115155 ± 70832	99626 ± 45297
*Ath32*	13	2.74	103.6	S1L134123348	71.6–117.6	LL	Additive	98310 ± 54368	121463 ± 72885	153891 ± 62109
** *Ath52* **	15	**5.42**	43.1	SSR151724023	34.2–99.0	LL	Additive	69437 ± 48256	128597 ± 63798	150222 ± 68151
** *Ath53* **	15	**4.17**	99.0	mUNC26203785	95.6–102.4	LL	Additive	91703 ± 61572	121455 ± 63792	156986 ± 69456

^a^ LOD scores were obtained from genome-wide QTL analysis using R/qtl. Significant QTL and LOD score were highlighted in bold. ^b^ 95% Confidence interval (CI) in Mb for significant or suggestive QTL estimated by R/qtl. ^c^ BB: homozygous for the BALB allele; LL: homozygous for the LP allele; H: Heterozygous for both BALB and LP alleles. Unit for carotid lesion: µm^2^. Allelic effects at each specific QTL were expressed as means ± SD.

**Table 2 cells-11-02669-t002:** Candidate genes for *Ath51* determined by haplotype analysis.

Chr	Position	Gene	dbSNP	Ref	BALB_cJ	LP_J	FVB_NJ	Csq	AA	AA Coord	SIFT
3	93323402	Hrnr	rs31474194	C	-	A	A	missense_variant	Q/K	316	**0.07**
3	93323777	**Hrnr**	rs214354165	T	-	A	A	missense_variant	S/T	441	**0.01**
3	93332492	Hrnr	rs33030796	G	-	C	C	missense_variant	G/R	3169	0.23
3	93332617	Hrnr	rs30076287	A	-	T	T	missense_variant	R/S	3210	**0**
3	93332685	Hrnr	rs30991551	G	-	A	A	missense_variant	R/K	3410	0.35
3	93471541	**Tchhl1**	rs33037297	G	-	T	T	missense_variant	E/D	517	**0.03**
3	93796525	Tdpoz4	rs243361353	T	-	C	C	missense_variant	L/P	43	0.51
3	94317417	Them4	rs30700506	C	-	T	T	missense_variant	R/C	34	0.21
3	94364230	C2cd4d	rs47866915	A	-	G	G	missense_variant	S/G	268	0.27
3	94488171	Celf3	rs33065732	C	-	T	T	missense_variant	P/S	291	0.25
3	94697430	Selenbp2	rs31664384	G	-	C	C	missense_variant	M/I	100	0.45
3	94761419	Cgn	rs387494032	C	-	T	T	missense_variant	D/N	1173	0.13
3	94770138	Cgn	rs47306886	C	-	T	T	missense_variant	V/M	693	0.16
3	96245528	Hist2h2aa2	rs31428119	G	-	T	t	5_prime_utr_variant	ref as H2ac19		
3	96269739	Hist2h2bb	rs30653282	G	-	A	A	5_prime_utr_variant	re as H2bc18	-	-
3	96285904	Fcgr1	rs31034407	G	-	A	A	missense_variant	A/V	259	0.22
3	96285911	Fcgr1	rs31666647	C	-	G	G	missense_variant	A/P	257	0.24
3	96292495	**Fcgr1**	rs51306537	G	-	A	A	missense_variant	A/V	23	**0**
3	96723591	Polr3c	rs31519177	C	-	G	G	missense_variant	R/P	45	0.28
3	98160914	Adam30	rs36577954	A	-	T	T	missense_variant	H/L	21	0.7
3	98713577	**Hsd3b2**	rs13477282	A	-	T	T	missense_variant	I/N	54	**0.02**
3	98716526	Hsd3b2	rs30748766	C	-	T	T	missense_variant	G/E	12	**0.01**
3	98716527	Hsd3b2	rs31516234	C	-	T	T	missense_variant	G/R	12	0.07
3	98806183	Hsd3b6	rs8245793	C	-	T	T	missense_variant	D/N	267	0.3
3	98880505	Hao2	rs33195243	A	-	T	T	missense_variant	S/T	203	0.42
3	99216497	**Wars2**	rs45786206	T	-	G	G	missense_variant	F/C	99	**0.02**
3	101815179	**Mab21l3**	rs47531228	C	-	T	T	missense_variant	R/H	377	**0.03**
3	101823288	Mab21l3	rs31248078	C	-	A	A	missense_variant	V/L	212	0.19
3	103171686	Bcas2	rs51633916	G	-	A	A	5_prime_utr_variant	-	-	-
3	103173240	**Bcas2**	rs48371516	T	-	C	C	missense_variant	S/P	65	**0**
3	104638700	Slc16a1	rs240468071	G	-	C	C	5_prime_utr_variant	-	-	-
3	104638781	Slc16a1	rs216996680	G	-	T	T	5_prime_utr_variant	-	-	-
3	104804630	Mov10	rs13459070	T	-	C	C	missense_variant	H/R	215	0.59
3	104825463	Capza1	rs46436792	T	-	C	C	missense_variant	I/V	222	0.32

Chr: chromosome; Position: in bp; dbSNP: Single nucleotide polymorphism database; Ref: Reference or C57BL/6J SNP; Csq: SNP consequences. AA: Amino acid; AA coord: Amino acid coordinate. Likely candidate genes and intolerant SNPs are highlighted in bold. “-” same as reference SNP. Some upstream variants are not shown due to space limitation.

## Data Availability

All data reported are included in Appendix A.

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
