# Peer review of "Phenotypic and Genetic Evidence for a More Prominent Role of Blood Glucose than Cholesterol in Atherosclerosis of Hyperlipidemic Mice"

_cells, 2022, doi:10.3390/cells11172669_

Round 1

Reviewer 1 Report

The manuscript by Abramson et al. identified that Ath51, Ath52, and Ath53 are associated with atherosclerosis lesions in hyperlipidemic mice models. The current study suggested a more prominent role for glucose than cholesterol in atherosclerotic plaque formation. I have several concerns discussed below:

•          What is the influence of the genetic background of the ApoE deficient mouse model on the current gene association study since only F2 mice were used?

•          In Fig.2, the fasting glucose level seems similar to the non-fasting glucose level in the mice. Please add the data that describe the current model by including all the baseline physiological/biochemical parameters.

•          Please show the histology images for the atherosclerosis lesion.

•          Are there any evidence proving that Ath51, Ath52, and Ath53 directly affect glucose metabolism?

Author Response

Reviewer # 1

The manuscript by Abramson et al. identified that Ath51, Ath52, and Ath53 are associated with atherosclerosis lesions in hyperlipidemic mice models. The current study suggested a more prominent role for glucose than cholesterol in atherosclerotic plaque formation. I have several concerns discussed below:

Comment: What is the influence of the genetic background of the ApoE deficient mouse model on the current gene association study since only F2 mice were used?

Response: F2 crosses are commonly used for QTL mapping when multiple phenotypes are analyzed in the same cross. The genome of each F2 mouse is the unique, random recombination of the genomes of two parental strains. As shown in Table 1, the alleles from LP mice increased atherosclerotic lesion sizes at QTL Ath32 on chromosome 13 and QTL Ath52 and Ath53 on chromosome 15 while the allele from BALB mice increased lesion sizes at QTL Ath51 on chromosome 3.    

Comment: In Fig.2, the fasting glucose level seems similar to the non-fasting glucose level in the mice. Please add the data that describe the current model by including all the baseline physiological/biochemical parameters.

Response: LOD score for non-fasting glucose is larger than the LOD score for fasting glucose. The biochemical characterization of plasma lipids and glucose in F2 mice has been reported (Int J Mol Sci 2022; 23: 6184).

Comment: Please show the histology images for the atherosclerosis lesion.

Response: Amended with the addition of figure 1.

Comment: Are there any evidence proving that Ath51, Ath52, and Ath53 directly affect glucose metabolism?

Response: The causal inference test can only be performed when QTL for two different traits are overlapping. After adjustment for atherosclerotic lesions, the peak LOD scores of Chr15 QTL for fasting and nonfasting glucose dropped (Fig. 6).  This further suggests gene connection between the two traits on Chr15.

Reviewer 2 Report

Overview and general recommendation:

I have read with interest the manuscript of Ashley M. Abramson, et al. This paper demonstrate genetic analysis of blood glucose and triglyceride with atherosclerosis using the intercross mice model between male LP-Apoe -/-mice and female BALB-Apoe -/-mice. Generally, my judgment is positive and I will recommend acceptance of the manuscript for publication, before which I recommend correcting some typos in the text.

1.    Can you please provide some theoretical explanation of mouse strains selection (the intercross between male LP-Apoe -/-mice and female BALB-Apoe -/-mice)? In ref 26, your groups demonstrated the variation in type 2 Diabetes-Related Phenotypes among Apolipoprotein E-Deficient Mouse Strains. C57BL/6, SWR/J, and SM/J Apoe-/- mice were susceptible to atherosclerosis and that C3H/HeJ and BALB/cJ Apoe-/- mice were relatively resistant. So why you choose LP/J?

2.     It will be better to delete loci names in Fig.1B, the font was too small for readers.

Author Response

Comment 1.    Can you please provide some theoretical explanation of mouse strains selection (the intercross between male LP-Apoe -/-mice and female BALB-Apoe -/-mice)? In ref 26, your groups demonstrated the variation in type 2 Diabetes-Related Phenotypes among Apolipoprotein E-Deficient Mouse Strains. C57BL/6, SWR/J, and SM/J Apoe-/- mice were susceptible to atherosclerosis and that C3H/HeJ and BALB/cJ Apoe-/- mice were relatively resistant. So why you choose LP/J?

Response: LP/J is among the 16 mouse strains whose genomes have been sequenced by Wellcome Sanger Institute (Nature Genetics 2018;50:1574–1583). When a QTL is found, sequence variant data allow for ready selection of underlying candidate genes.

Comment 2: It will be better to delete loci names in Fig.1B, the font was too small for readers.

Response: Fig. 1B (now Fig. 2B) was plotted by QTX software.  Its plotting function is hardly manageable.

Reviewer 3 Report

Introduction

The introduction is quite fragmented, lacks a bit of flow and cohesiveness. It reads like a ‘list’ of statements which are not put in context or for which the transitions are missing.

There are parts of the introductions for which a different font style has been used.

It feels like the authors ‘threw in’ a bunch of ‘fancy’ words to make their writing sound better than it is. However ,these words don’t necessarily add anything of substance.

Materials and methods

Why did the authors not get a baseline value of the plasma/blood? How can they be sure that the values they are reporting are due to the diet? These results could be due to the specific mouse strain…

Results

There are things mentioned in this section which were never introduced in the introduction.

The results are primarily based on genomic data. However, what is important from a disease perspective is what happens on protein level. Consequently, this manuscript would have benefited tremendously by a verification of the reported values through the means of confocal microscopy (immunofluorescence) or by immunoblotting.

The correlation of lesion size with plasma glucose values/HDL/triglycerides is based on a one-time measurement. Physiologically, the atherogenesis is caused by long-term ‘exposure’ to these risk factors. Consequently, these correlations are not that convincing… The attempt of correlate atherogenesis with various parameters without introducing these or without explaining this correlation is also a bit puzzling.

The entire results section is based on a lot of assumptions and predictions. A fact which is also highlighted in the writing.

Author Response

Comment on Introduction: The introduction is quite fragmented, lacks a bit of flow and cohesiveness. It reads like a ‘list’ of statements which are not put in context or for which the transitions are missing.

There are parts of the introductions for which a different font style has been used.

It feels like the authors ‘threw in’ a bunch of ‘fancy’ words to make their writing sound better than it is. However ,these words don’t necessarily add anything of substance.

Response: We agree with the reviewer’s comments and have had this portion reworked. 

Comment on Materials and methods: Why did the authors not get a baseline value of the plasma/blood? How can they be sure that the values they are reporting are due to the diet? These results could be due to the specific mouse strain…

Response: The baseline values of plasma lipids and glucose levels before mice started the Western diet are included in Supplemental materials.  These data are not presented due to limited relevance to the current topic. The dramatic impact of Western diet on plasma lipid and glucose levels of Apoe-/- mouse strains has been reported (PLoS One 2015;10:e0120935. Cardiovasc Diabetol 2011;10:117).  QTL analysis is based on mouse strain differences in phenotypes to find underlying causal genes.  

Comment on Results: There are things mentioned in this section which were never introduced in the introduction.

The results are primarily based on genomic data. However, what is important from a disease perspective is what happens on protein level. Consequently, this manuscript would have benefited tremendously by a verification of the reported values through the means of confocal microscopy (immunofluorescence) or by immunoblotting.

The correlation of lesion size with plasma glucose values/HDL/triglycerides is based on a one-time measurement. Physiologically, the atherogenesis is caused by long-term ‘exposure’ to these risk factors. Consequently, these correlations are not that convincing… The attempt of correlate atherogenesis with various parameters without introducing these or without explaining this correlation is also a bit puzzling.

The entire results section is based on a lot of assumptions and predictions. A fact which is also highlighted in the writing.

Response: The introduction has reworked to clarify the rationale for this study.

The methods for lipid and glucose measurements are widely used in research and clinical labs. 

The correlations of lesion sizes with plasma glucose values/HDL/triglycerides are determined under both fasting (Fig. 4 A to D) and nonfating states (Fig. 4 E to H).

The use of high density SNP markers allowed us to directly localize QTL for each trait besides inferring genotypes between markers through interval mapping.

Reviewer 4 Report

The manuscript demonstrated a more prominent role for blood glucose than cholesterol in atherosclerotic plaque formation of hyperlipidemic mice.

My commment

 Is there a role of insulin resistance? HOMA-IR results should be included and discussed.

Below are additional comments 1-Figure 1: add scale bar 2-In conclusion, the authors should indicate therapeutic and/or diagnostic impact of the study 3-In materials and methods: plasma levels of TG, HDL, non-HDL, TGs should be separated under different title

Author Response

Commment: “Is there a role of insulin resistance? HOMA-IR results should be included and discussed”.

Response: Plasma insulin was not measured for the current cross, and thus HOMA-IR and the contribution of insulin resistance to diabetes etiology could not been determined.  However, we previously have demonstrated that impaired insulin secretion rather than insulin resistance underlies the development of western diet-induced type 2 diabetes in Apoe-/- mouse strains (Cardiovasc Diabetol 2011;10:117. PLoS One 2015;10:e0120935).  In a segregating F2 population derived from Apoe-/- mouse strains and fed a Western diet, we found that plasma insulin levels are poorly correlated with plasma glucose levels (Hum Mol Genet 2006;15:1650).  We have addressed this issue in Discussion.   

Comment 1-Figure 1: add scale bar

Response: Amended.

Comment 2-In conclusion, the authors should indicate therapeutic and/or diagnostic impact of the study

Response: Amended.

Comment 3-In materials and methods: plasma levels of TG, HDL, non-HDL, TGs should be separated under different title

Response: Revised.

Reviewer 5 Report

The authors presented the study very well and the experiments are very appropriate and detailed. The results demonstrate phenotypic and genetic associations of blood glucose and triglyceride with atherosclerosis, and suggest a more prominent role for blood glucose than cholesterol in atherosclerotic plaque formation in hyperlipidemic mice.  I found rigorous and correct the methodology of selection and evaluation of data here presentedHowever, I believe that it requires a major revision to assess several flaws.

Minor criticisms:

1) Only male mice were studied. Please provide a brief justification why this specific sex was studied in the "Methods section".

2) Have been the analysis corrected for biological variable such as BMI? BMI is an important factor relevant in atherosclerosis.

Author Response

Comments and Suggestions for Authors

The authors presented the study very well and the experiments are very appropriate and detailed. The results demonstrate phenotypic and genetic associations of blood glucose and triglyceride with atherosclerosis, and suggest a more prominent role for blood glucose than cholesterol in atherosclerotic plaque formation in hyperlipidemic mice.  I found rigorous and correct the methodology of selection and evaluation of data here presented. However, I believe that it requires a major revision to assess several flaws.

Minor criticisms:

Comment 1: Only male mice were studied. Please provide a brief justification why this specific sex was studied in the "Methods section".

Response: Amended.

Comment 2: Have been the analysis corrected for biological variable such as BMI? BMI is an important factor relevant in atherosclerosis.

Response: We measured body weight but not body length so BMI could not be calculated.  Our magnetic resonance imaging study of a congenic strain and its background strain has shown that visceral and subcutaneous fat rather than non-fat tissues accounts for their variation in body weight (PLoS One 2018;13:e0204071).  We have addressed this issue in Discussion.

Round 2

Reviewer 1 Report

The authors answered my concerns, and I don't have further questions.  

Author Response

We thank the reviewer for your input.

Reviewer 3 Report

The authors have addressed this reviewer's previous concerns

Author Response

We thank the reviewer for your time and effort.

Reviewer 4 Report

The authors addressed my comments